# An Empirical Study on the Effect of Quick and Careful Labeling Styles in Image Annotation

Chia-Ming Chang*
The University of Tokyo

Xi Yang†
Jilin University

Takeo Igarashi‡
The University of Tokyo

## ABSTRACT

Assigning a label to difficult data requires a long time, particularly when non-expert annotators attempt to select the best possible label. However, there have been no detailed studies exploring a label selection style during annotation. This is very important and may affect the efficiency and quality of annotation. In this study, we explored the effects of labeling style on data annotation and machine learning. We conducted an empirical study comparing "quick labeling" and "careful labeling" styles in image-labeling tasks with three levels of difficulty. Additionally, we performed a machine learning experiment using labeled images from the two labeling styles. The results indicated that quick and careful labeling styles have both advantages and disadvantages in terms of annotation efficiency, label quality, and machine learning performance. Specifically, careful labeling improves label accuracy when the task is moderately difficult, whereas it is time-consuming when the task is easy or extremely difficult.

**Keywords**: Cognitive Psychology, Labeling Style, Non-Expert Data Annotation, Data Collection, Machine Learning.

**Index Terms**: • Computing methodologies~Artificial intelligence ~Philosophical/theoretical foundations of artificial intelligence ~Cognitive science

## 1 INTRODUCTION

A large, high-quality dataset is necessary to obtain better machine learning results. However, it is expensive to recruit a large number of expert annotators (who have sufficient domain knowledge) to work on it. Recruiting non-expert annotators (typically crowd workers) is cheaper and easier; therefore, it is often the only viable option in practice [20, 21]. However, label quality is critical in non-expert data annotation (i.e., crowdsourcing tasks) [22, 23, 35]. Various annotation methods and tools have been introduced to address this issue [13, 14, 29, 32, 40]. However, there are no detailed studies examining this issue from a human perspective (i.e., cognitive psychology), such as investigating the effect of a label selection style during annotation (i.e., how a user makes a label decision). This is important, and it could affect annotation efficiency and quality. The different labeling styles used by annotators could affect the annotation efficiency and quality.

*e-mail: chiaming@ui.is.s.u-tokyo.ac.jp
†e-mail: yangxi21@jlu.edu.cn
‡e-mail: takeo@acm.com

This study presents an empirical study comparing two labeling styles (quick labeling and careful labeling) for a manual image labeling task with three datasets under different levels of data difficulty: easy (MNIST), moderately difficult (Fashion-MNIST), and extremely difficult (Kuzushiji-MNIST). Thereafter, we conducted a machine learning experiment using the labeled images with quick labeling and careful labeling styles and compared the machine learning results (classification accuracy), as shown in Figure 1.

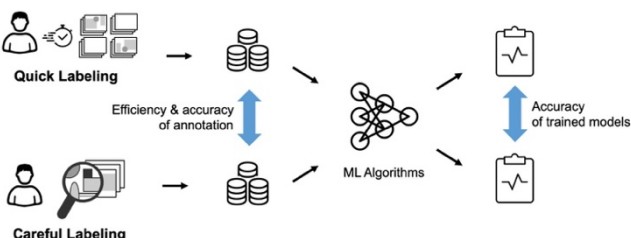

Figure 1: Research Overview.

These results indicated that the labeling style affects annotation efficiency (task completion time) and label accuracy. The careful labeling style exhibited significantly higher accuracy than the quick labeling style when the dataset was moderately difficult. Moreover, the results of the machine learning experiment indicated that the labeled images (training data) collected via the careful labeling style could achieve better machine learning performance (higher classification accuracy) than that collected via the quick labeling style. However, the careful labeling style did not bring benefits when the images were easy (i.e., label accuracy was already high via the quick labeling style) or extremely difficult (i.e., the careful labeling style could not significantly improve the label accuracy). We discussed the effects of the two labeling styles according to three different levels of data difficulty (easy, moderately difficult, and extremely difficult), and we discussed three factors that need to be carefully considered when selecting an appropriate labeling style for an annotation task. This study makes the following contributions:

- Identifying labeling styles as a variable in non-expert image annotation and machine learning.
- An empirical study comparing quick labeling and careful labeling styles, demonstrated the benefits of the careful labeling style in an image annotation task.
- A machine learning experiment using labeled images with different labeling styles, demonstrated the effects of the labeling styles on image classification accuracy.

## 2 RELATED WORK

### 2.1 Manual Data Annotation and Challenges

Manual data annotation is a basic practice in machine learning. A large size is often necessary to improve machine learning results. Popular datasets including ImageNet [28], AudioSet [30], and YouTube-8M [31], which were manually labeled by human annotators, were applied. The manual data annotation process is extremely tedious and time-consuming, and many studies have proposed annotation tools for assisting manual data annotation. For instance, LabelMe [10] is a web-based image annotation tool that allows multiple annotators to label an image and share their labeling results instantly. ESP [11] is an image annotation tool combined with a computer game that provides an enjoyable labeling process for annotators, and TagATune [27] is an audio annotation tool that shares the same idea. VIA [12] is an annotation tool that allows annotators to define and describe spatial regions in images, audio segments, and video frames. iVAT [32] is a video annotation tool that supports manual, semi-automatic, and automatic video annotation.

Most of these tools provide supportive, efficient, and enjoyable systems for improving the tedious processes of manual data annotation. These tools generally assume that annotators have sufficient domain knowledge for labeling tasks. However, data annotation tasks often rely on non-expert annotators who lack sufficient domain knowledge because access to a sufficient number of expert annotators is limited and expensive [18, 19]. Therefore, labeling tasks can be significantly difficult for non-expert annotators and the labeled data may contain numerous errors [23] [33] [36]. However, these annotation tools may not be able to address this issue when annotators are non-experts.

### 2.2 Annotation Workflows for Improving the Label Quality

Many data annotation workflows have been proposed to improve the label quality, particularly for annotation tasks conducted through crowdsourcing. Revolt [13] is a collaborative crowdsourcing labeling workflow that applies concepts from expert annotation workflows (label-check-modification). This specific workflow can produce higher label quality than a conventional labeling workflow. Pairwise HITS [14] is a labeling workflow for quality estimation that allows annotators to compare a pair of labeled data and select the better one. Fang et al. [32] introduced a two-round workflow to improve the quality of crowdsourced image labeling. During the first round, the annotators select a label for the target images (several labels are assigned to each image). During the second round, the annotators are required to select the best label for each image (referring to the results from other annotators). Baba [29] introduced two types of labeling workflows (parallel and interactive) that allow multiple annotators to be involved in an annotation task in different ways to improve the label quality. In addition, various studies have used the concept of hierarchical classification in data annotation to increase the labeling efficiency and label quality [15] [16].

The main concept of these annotation workflows is the gathering of knowledge from multiple individuals. This is a typical workflow for improving the label quality by involving a group of annotators (non-experts or experts) to collaborate for a labeling task. In this study, we aim to explore the "labeling styles" rather than the "labeling workflows" during a data annotation task. In addition to machine learning, data annotation has been used in various research areas. For instance, social scientists annotate data to discover interesting phenomena and establish theories [7] [9], and data annotations, such as a thematic analysis approach, is often used to analyze qualitative data [8]. Data annotation is not only a labeling process but also a cognitive process by which annotators view data, organize concepts, and make labeling decisions. Concept organization plays a crucial role in data annotation. Kulesza et al. [17] indicated that annotators often organize their conceptual similarity by observing more items during data annotation. Chang et al. [40] shared the same concept and proposed a spatial layout labeling interface for concept organization during the annotation process. We believe that cognitive processes (i.e., how to make a labeling decision) in a manual data annotation task are important and they could affect the label quality and cost.

### 2.3 Intuitive and Systematic Decision-Making

Cognitive style (or thinking style) is a term used in cognitive psychology to describe ways in which individuals organize and process information, and finally make decisions [1] [2] [3]. Intuitive and systematic decision-making are two types of cognitive style. Intuitive decision-making is a type of associative thinking that relies on intuition, and systematic decision-making is a type of rule-based thinking that relies on logical evaluation [4]. Both cognitive styles have advantages and disadvantages. For instance, intuitive decision-making requires less time than systematic decision-making. However, systematic decision-making involves a deeper consideration process than intuitive decision-making.

These two cognitive styles have been used and discussed in several fields. Sagiv et al. [3] analyzed different intuitive and systematic cognitive styles used by art, accounting, and mathematics students. They established that different students prefer different cognitive styles in their class, and the cognitive style is consistent with an individual's personal attributes. Ma-Kellams and Lerner [5] compared intuitive and systematic cognitive styles to understand the feelings of other people, and they established that a systematic cognitive style can produce better empathic accuracy than an intuitive cognitive style. Hwang and Lee [6] explored the impact of intuitive and systematic cognitive styles of consumers on their visual attention patterns in online shopping environments. The results indicated that consumers pay different visual attention to webpages when they use different cognitive styles to make purchasing decisions. These studies have shown the effects of the cognitive styles used in various activities. In this study, we share a similar concept to explore the effects of cognitive styles (labeling styles) in manual data annotation, where annotators complete a labeling task (i.e., select an appropriate label for an image) through intuitive (quick labeling) and systematic decision-making (careful labeling).

## 3 LABELING STYLE AND USER INTERFACE

We defined two labeling styles of manual image annotation based on the theory of cognitive psychology: quick labeling and careful labeling.

**Quick labeling.** Quick labeling refers to intuitive decision-making. Here, annotators select a label for an image as quickly as possible even when they lack confidence in the target image and label. They are strongly encouraged to select labels within a short time.

**Careful labeling.** Careful labeling refers to the concept of systematic decision-making. Here, annotators select an image label as carefully as possible, particularly when they are not confident of the target image and label. They are strongly encouraged to spend sufficient time before making a label decision.

A labeling system was developed to evaluate the quick labeling and careful labeling styles in a manual image annotation task. Figure 2 shows a screenshot of the image-labeling interface. The left side of the interface lists the labels (10 labels/categories in the Fashion-MNIST dataset), and the right side of the interface

represents the target image. During the labeling task, the annotators were asked to select an appropriate label from the label list and apply it to the target image. After selecting a label for an image (by clicking on a label), the system automatically moves to the next image. The annotators were not allowed to return to the previous images after selecting an image label.

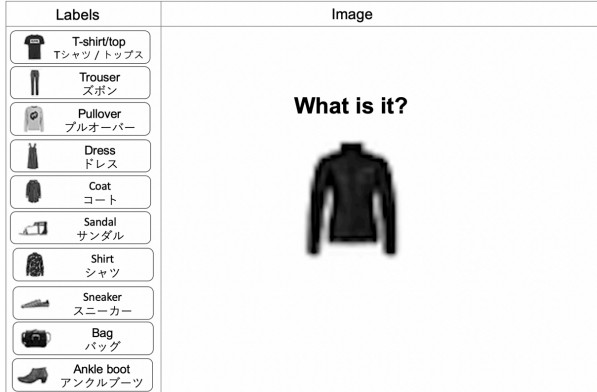

Figure 2:   Screenshot of the labeling interface.

## 4   USER STUDY

A user study was conducted to compare the quick labeling and careful labeling styles applied to an image-labeling task. We aimed to observe the effects of the two labeling styles in image labeling tasks, specifically at the three different levels of data difficulty (easy, moderately difficult, and extremely difficult). We compared quick labeling and careful labeling styles in terms of label accuracy and labeling time for the given image labeling tasks.

### 4.1   Apparatus

To control user study quality, specifically the use of quick labeling and careful labeling styles during image-labeling tasks, we outsourced the execution of the user study to a professional company, which asked their employees to participate in the user evaluation process as part of their job. The total cost was approximately $1600, that is, $640 for the quick labeling task ($53 per participant) and $960 for the careful labeling task ($80 per participant). During the user study, the participants were asked to sit in front of a desktop and complete the given image-labeling tasks (Figure 3).

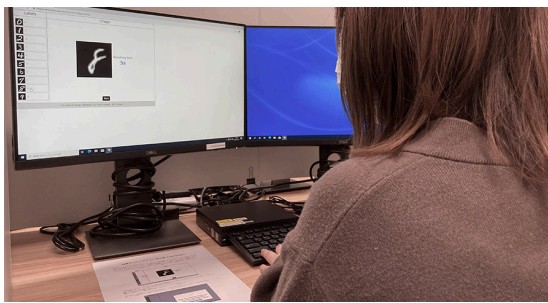

Figure 3:   Photography of user study.

### 4.2   Participants

Twenty-four participants (12 men and 12 women, 18–49 years) were invited by the company to participate in the user study. All the participants were Japanese (i.e., understood Japanese Hiragana letters). Most of the participants (n = 19) had no prior experience

with data annotation, four had less than half a year of experience, and one had between half and a full year of experience.

### 4.3   Dataset

Three datasets, MNIST [24], Fashion-MNIST [25], and Kuzushiji-MNIST [26], were used for labeling tasks in the user study. Each dataset contained 60,000 training images and 10,000 testing images in ten categories (labels). Figure 4 shows the ten categories for each dataset.

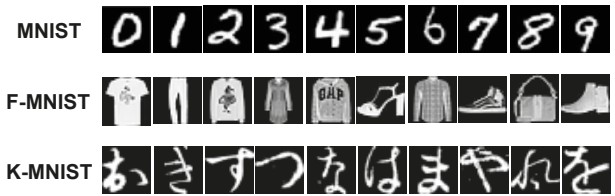

Figure 4:   Ten categories in the MNIST, F-MNIST, and K-MNIST datasets.

The datasets had varying levels of difficulty. MNIST is an "easy" dataset. These handwritten number digits are not difficult for human users to recognize, even when they are in difficult cases, as shown in Figure 5(a). Fashion-MNIST is a "moderately difficult" dataset. It is because it contains some difficult (confusing) items (e.g., "Pullover" and "T-shirt/Top"). Figure 5(b) shows examples of easy and difficult (confusing) cases. Kuzushiji-MNIST is an "extremely difficult" dataset. The handwritten Japanese Hiragana letters are very difficult to recognize (even for Japanese users), specifically when the letters are in difficult cases. Figure 5(c) shows examples of easy and difficult cases.

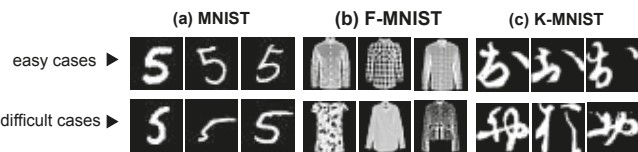

Figure 5:   Examples of easy (upper part) and difficult (lower part) cases from the MINIST, F-MNIST and K-MNIST datasets.

We randomly selected 100 images (10 images per label) from each dataset. Thereafter, we created 12 100-image datasets (e.g., Datasets 1–12) with no overlapping images (1200 images from each dataset, and ten split into 12 non-overlapping subsets). The 12 100-image datasets were used for 12 participants in the quick labeling task and 12 participants in the careful labeling task (the same datasets were used for both labeling styles).

### 4.4   Task and Condition

The image labeling tasks involved labeling 300 images (100 images for each dataset) for each participant. During the image labeling task, the participants were requested to select an appropriate label from a 10-category list (10 labels) for each image. A between-subject method was used, in which 12 of the participants were asked to complete the labeling task using the quick labeling style, and the other 12 participants were asked to complete the labeling task using the careful labeling style.

**Quick Labeling Conditions.** The participants of the quick labeling task were provided with the following instructions for the labeling task:

   *"Please select a label for an image as quickly as possible. Here, if you are unsure about the target images and labels, please*

*simply select the most appropriate one (i.e., make a guess based on your intuition). Please do not spend too much time considering this before selecting the label. We STRONGLY ENCOURAGE you to select a label for an image within 5 s. A 5 s-timer is provided on the labeling interface. After selecting a label for an image, the system will automatically show the next image. This means that you are not allowed to change your selected label."*

Our labeling system for the label-quick task contains a 5 s-timer for labeling each image. An alert message "Time's up! Please select a label now" is displayed when the timer ends, as shown in Figure 6.

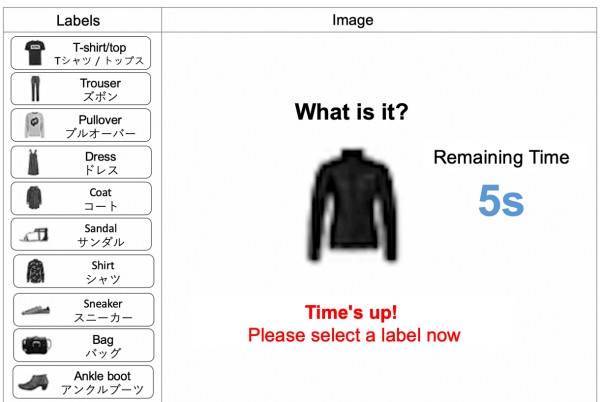

Figure 6: Timer and alert message in the labeling system.

The system did not automatically move to the next image until the participants selected a label for the target image, even when the timer ended. The timer was designed as a reminder to the participants during annotation. In addition, participants were not allowed to return to the previous image after selecting a label for the image.

**Careful Labeling Conditions.** The participants in the careful labeling task were provided with the following instructions:

*"Please select a label for an image as carefully as possible. There is no time limitation for labeling each image. You have sufficient time to think carefully before making a label decision, particularly when the target images are difficult or when you are not confident about the images and labels. We STRONGLY ENCOURAGE you to spend sufficient time (there is no time limitation) before making a label decision. After selecting a label for an image, the system will automatically show the next image. This means that you are not allowed to change your selected label."*

The labeling system for the careful labeling task is the same as that used for the quick labeling task; but there is no timer shown on the labeling interface (Figure 2). The participants were allowed to spend as much time as necessary to select a label. In addition, the participants were not allowed to return to the previous image after selecting a label for an image. The participants were informed that they could receive 1.5 times higher rewards for the careful labeling task than for a normal labeling task (i.e., quick labeling task).

### 4.5 Procedure

The instructor provided an oral overview and detailed written instructions to the participants. The evaluation itself was composed of three parts (in order): instruction and trial (5–10 min), labeling tasks (20–45 min), and questionnaire (3–5 min).

The entire evaluation process was completed within 40–60 min (depending on the labeling style). After providing instructions on the labeling interfaces and the given tasks, the participants were allowed to practice on a small labeling task (to label three images for each dataset that differed from the images used in the formal tasks) before starting the given image labeling tasks.

### 4.6 Measurement

Our labeling system automatically recorded and measured the time and accuracy of the image-labeling tasks completed by the participants. The timer started when the participants clicked on "START" and stopped when they clicked on "FINISH." In addition, the system recorded the time spent by the participants for each image-labeling process. After the image labeling tasks, the participants were asked to answer a questionnaire regarding the labeling process. The questionnaire contained three Likert-scale questions for each labeling style (Section 6.4).

## 5  MACHINE LEARNING EXPERIMENT

In the user study, 7200 labeled images were collected (2400 for each dataset and 1200 for each labeling style in a dataset). The training dataset contained errors made by the participants (i.e., the accuracy rate of the training data was not 100%). We used them as the training data to perform a machine learning experiment to evaluate the effects of labeling styles (data collected via the quick labeling and careful labeling styles) on machine learning accuracy (image classification). Three common machine learning algorithms (logistic regression, K-nearest neighbors, and support vector machine) were selected for the case study in the machine learning experiment. We did not use more advanced techniques (e.g., deep learning) because the training dataset was too small and our goal was not to pursue high machine learning accuracy but to compare the difference between the two labeling styles. The testing data used in the machine learning experiment were 30,000 images (10,000 for each dataset), which were different from the training dataset (7200 labeled images).

## 6  RESULTS

### 6.1  Task Completion Time

Figure 7 shows the task completion times for the different labeling styles and datasets. The results from the quick labeling task indicated that the participants spent an average of 4 min and 23 s, 5 min and 6 s, as well as 5 min and 37 s to label the 100 images with the MNIST, Fashion-MNIST, and Kuzushiji-MNIST datasets, respectively. The results indicated that the participants spent an average of 4 min and 42 s, 6 min and 58 s, and 10 min and 13 s labeling the 100 images in the careful labeling task. The results of an unpaired t-test of the task completion time indicated that the difference was insignificant ($p > 0.05$) between the quick labeling and careful labeling styles in the MNIST dataset, whereas there were significant differences ($p < 0.01$) in the Fashion-MNIST and Kuzushiji-MNIST datasets. This indicates that the careful labeling style requires a longer time to complete labeling tasks than the quick labeling style when the images are moderately and extremely difficult. However, the task completion time was comparable between the quick and careful labeling styles when the images were easy.

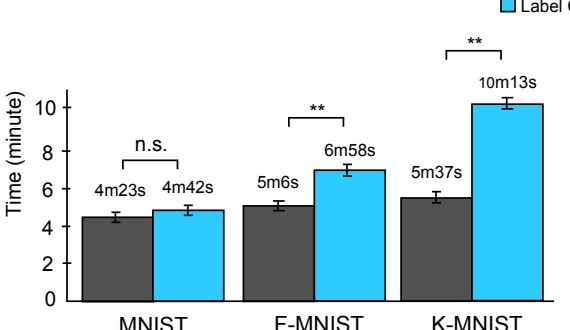

Figure 7: Task completion time. MNIST. LQ: mean = 4.23; SD = 0.73; LC: mean = 4.42; SD = 0.61. F-MNIST. LQ: mean = 5.06; SD = 0.76; LC: mean = 6.58; SD = 1.13. K-MNIST. LQ: mean = 5.37; SD = 0.90; LC: mean = 10.13; SD = 3.55.

## 6.2 Annotation Accuracy

Figure 8 shows the accuracy of the labels given by the participants in the quick labeling and careful labeling styles. The results from the quick labeling task indicated accuracies of 97.58%, 72.08%, and 58.08% for the three datasets, and the results from the careful labeling tasks indicated accuracies of 97.58%, 76.83%, and 60.08%. The analysis of accuracy using an unpaired t-test showed that the difference was insignificant ($p > 0.05$) in the MNIST and Kuzushiji-MNIST datasets, whereas there was a significant difference ($p < 0.05$) between the two labeling styles in the Fashion-MNIST dataset. This indicates that the careful labeling style can help non-expert annotators to select labels more correctly when the images are moderately difficult (Fashion-MNIST), whereas no clear benefit was observed when the images were easy (MNIST) and extremely difficult (Kuzushiji-MNIST). This indicates that conducting a careful labeling task is expensive when the task is either easy or difficult.

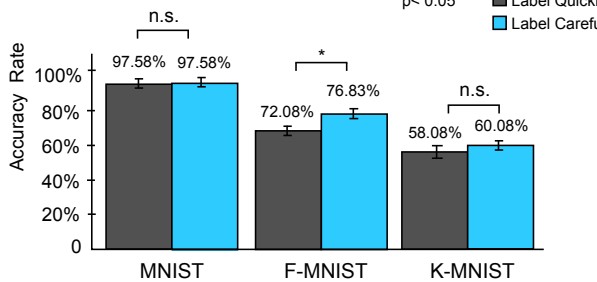

Figure 8: Accuracy of Labeling Tasks. MNIST. LQ: mean = 97.58; SD = 1.44; LC: mean = 97.58; SD = 1.56. F-MNIST. LQ: mean = 72.08; SD = 4.17; LC: mean = 76.83; SD = 6.64. K-MNIST. LQ: mean = 58.08; SD = 6.53; LC: mean = 60.08; SD = 7.82.

## 6.3 Temporal Effect

*Task Completion Time*

Figure 9 shows the average time for the labeling process for the first half (1–50 images) and second half (51–100 images) in the three datasets using the quick labeling style. The results indicated that the participants spent an average of 2 min 6 s and 2 min 17 s, 2 min 36 s and 2 min 30 s, and 2 min 29 s and 3 min 8 s to complete the first and second halves of the MNIST, Fashion-MNIST, and Kuzushiji-MNIST datasets, respectively. The results of the paired t-test indicated that the difference was not significant ($p > 0.05$) between the first and second halves of the labeling process in the MNIST and Fashion-MNIST datasets, but the difference was significant ($p < 0.05$) in the Kuzushiji-MNIST

dataset. This indicates that there is a temporal effect at all levels of data difficulty when using the quick labeling style. However, the results interestingly indicated that the participants spent a longer time completing the second half of the image-labeling task.

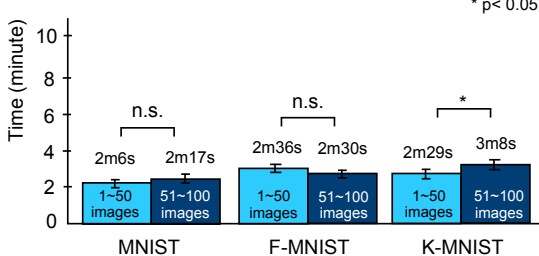

Figure 9: Average time of the first and second 50 images in the quick labeling style. MNIST (1–50 images): mean = 2.06; SD = 0.38; MNIST (51–100 images): mean = 2.17; SD = 0.35. F-MNIST (1–50 images): mean = 2.36; SD = 0.28; F-MNIST (51–100 images): mean = 2.30; SD = 0.48. K-MNIST (1–50 images): mean = 2.29; SD = 0.21; K-MNIST (51–100 images): mean = 3.08; SD = 0.27.

Figure 10 shows the average time for the labeling process for the first half (1–50 images) and second half (51–100 images) using the careful labeling style in different datasets. The results indicated that the participants spent an average of 2 min 24 s and 2 min 18 s, 4 min 5 s and 2 min 53 s, and 4 min 21 s and 5 min 52 s to complete the first and second halves of the MNIST, Fashion-MNIST, and Kuzushiji-MNIST datasets, respectively. The results of the paired t-test indicated that the difference was not significant ($p > 0.05$) between the first and second halves of the labeling process in the MNIST dataset, but the difference was significant ($p < 0.05$) in the Fashion-MNIST and Kuzushiji-MNIST datasets. This indicates that there is no temporal effect in the use of the careful labeling style when the images are easy. However, a temporal effect was observed when the images were moderately difficult. The participants significantly increased their labeling speed in the second half of the image-labeling task. In addition, the participants spent more time in the second half when the images were extremely difficult, which is the same as the quick labeling style.

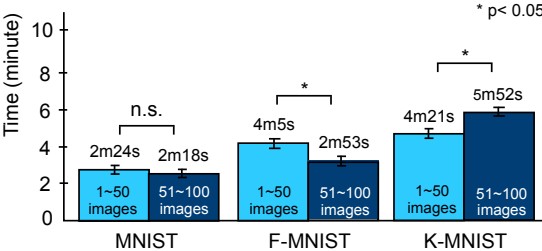

Figure 10: Average time of the first and second 50 images in the careful labeling style. MNIST (1–50 images): mean = 2.24; SD = 0.33; MNIST (51–100 images): mean = 2.18; SD = 0.29. F-MNIST (1–50 images): mean = 4.05; SD = 0.57; F-MNIST (51–100 images): mean = 2.53; SD = 0.38. K-MNIST (1–50 images): mean = 4.21; SD = 1.02; K-MNIST (51–100 images): mean = 5.52; SD = 0.73.

*Annotation Accuracy*

Figure 11 shows the accuracy of the labeling process in the first half (1–50 images) and the second half (51–100 images) via the quick labeling style. The results indicated that the accuracy rates were 97.5% and 97.67% in the first and second halves of the

MNIST dataset, 73.17% and 71% in the Fashion-MNIST dataset, as well as 56.67% and 59.5% in the Kuzushiji-MNIST dataset. The results of the paired t-test indicated that the difference was not significant (p > 0.05) between the first and second halves of the labeling process for all datasets. This indicates that there is no temporal effect at any level of data difficulty when using the quick labeling style. The label accuracy was not significantly affected (improved) by different labeling styles.

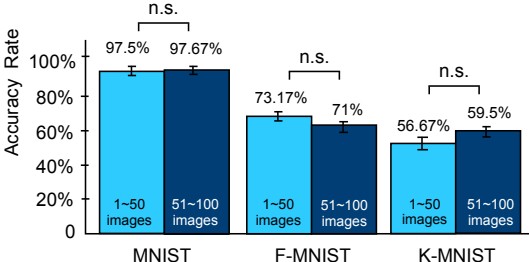

Figure 11: Average time of the first and second 50 images in the quick labeling style. MNIST (1–50 images): mean = 97.50; SD = 1.27; MNIST (51–100 images): mean = 97.67; SD = 1.53. F-MNIST (1–50 images): mean = 73.17, SD = 3.93; F-MNIST (51–100 images): mean = 71, SD = 4.31. K-MNIST (1–50 images): mean = 56.67; SD = 6.95; K-MNIST (51–100 images): mean = 59.50; SD = 5.47.

Figure 12 shows the accuracy of the labeling process in the first half (1–50 images) and the second half (51–100 images) via the careful labeling style. The results indicated that the accuracy rates were 97.17% and 98% in the first and second halves of the MNIST dataset, 73.67% and 80% in the Fashion-MNIST dataset, as well as 62.83% and 57.33% in the Kuzushiji-MNIST dataset. The results of the paired t-test indicated that the difference was not significant (p > 0.05) between the first and second halves of the labeling process for the MNIST and Kuzushiji-MNIST datasets. However, the difference was significant (p < 0.05) for the Fashion-MNIST dataset. This indicates that there is no temporal effect when the images are easy and extremely difficult to use with the careful labeling style. However, a temporal effect was observed when the images were moderately difficult. The participants could significantly improve the label accuracy in the second half of the image-labeling task.

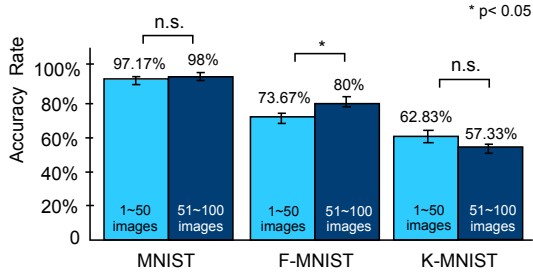

Figure 12: Average time of the first and second 50 images in the careful labeling style. MNIST (1–50 images): mean = 97.17; SD = 1.83; MNIST (51–100 images): mean = 9; SD = 1.35. F-MNIST (1–50 images): mean = 73.67; SD = 4.95; F-MNIST (51–100 images): mean = 80, SD = 6.58. K-MNIST (1–50 images): mean = 62.83; SD = 8.13; K-MNIST (51–100 images): mean = 57.22; SD = 7.22.

In summary, the careful labeling style has a temporal effect during the labeling process in the task completion time and accuracy rate (i.e., reduced time and increased accuracy) only

when the images are moderately difficult. When the images are too easy, there is no temporal effect. Interestingly, if the images are too difficult, the task completion time is longer in the second half.

### 6.4 Questionnaire

Figure 13 shows how confident the participants felt in the given image labeling tasks. In the MNIST dataset, the results indicated that most of the participants felt extremely confident or confident when selecting a label for an image using either the quick labeling (n = 10) or careful labeling (n = 9) styles, whereas none of the participants felt apprehensive. For the Fashion-MNIST dataset, the results indicated that only one participant felt extremely confident when selecting a label for an image through the quick labeling style and only two participants felt confident when selecting a label for an image through the careful labeling style. In the Kuzushiji-MNIST dataset, the results indicated that no participants felt confident or extremely confident when selecting a label for an image through either the quick labeling or careful labeling styles. More participants using the careful labeling style felt apprehensive (n = 4) or extremely apprehensive (n = 7) in comparison to the participants using the quick labeling style (apprehensive, n = 3; extremely apprehensive, n = 6). This indicates that the labeling styles do not affect the subjective impression of the participants' confidence during annotation. However, the ambiguities in the data affect the confidence of the participants in selecting a label for an image during annotation.

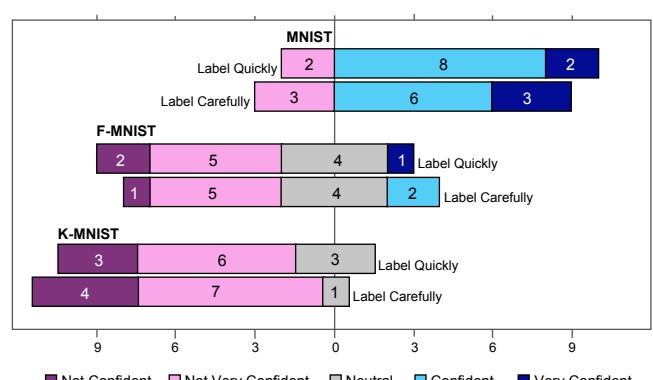

Figure 13: Confidence of the participants when selecting a label from the MNIST, F-MNIST, and K-MNIST datasets.

### 6.5 Results of Machine Learning Experiment

Figure 14 presents the machine learning results (i.e., image classification accuracy) for the three datasets with the quick and careful labeling styles.

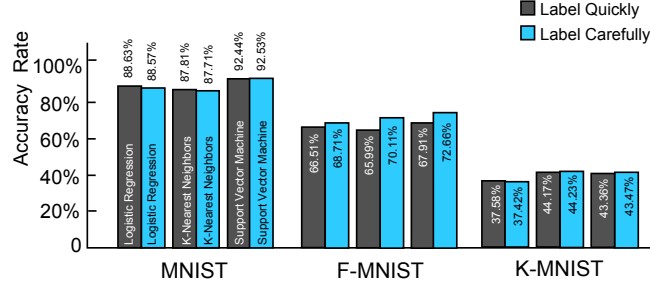

Figure 14: Accuracy of machine learning models in the MNIST, F-MNIST, and K-MNIST datasets.

In the data analysis, we did not compute the accuracy for each participant because the dataset was too small. We combined all annotations, trained the model, and measured the performance of the model.

**MNIST Dataset.** The accuracy of the training data was 97.58% for both the quick labeling and careful labeling styles (Figure 12). Based on the accuracy of the training data, the machine learning performance (accuracy) showed almost no differences between the two labeling styles (logistic regression, LQ = 88.63%, LC = 88.57%; K-nearest neighbors, LQ = 87.81%, LC = 87.71%; support vector machine, LQ = 92.44%, LC = 92.53%). These results were not surprising because the label accuracy of the training data was the same.

**Fashion-MNIST Dataset.** The accuracy of the training data was 72.08% for the quick labeling style and 76.83% for the careful labeling style (Figure 12), which is a significant difference between the two labeling styles. Based on the accuracy, the machine learning performance (accuracy) showed that there were differences between the two labeling styles (logistic regression, LQ = 66.51%, LC = 68.71%; K-nearest neighbors, LQ = 65.99%, LC = 70.11%; support vector machine: LQ = 67.91%, LC = 72.66%). The differences were between 2.2% and 4.12%. Machine learning algorithms often work well even if the labels given to difficult data contain errors. Our results indicate that improving the label accuracy via the careful labeling style can also improve the accuracy of machine learning.

**Kuzushiji-MNIST Dataset.** The accuracy of the training data was 58.08% for the quick labeling style and 60.08% for the careful labeling style (Figure 12). There was a small difference of 2% between the two labeling styles, but it was not significant according to the paired t-test analysis. Based on the accuracy of the training data, the machine learning performance (accuracy) showed that the differences were significantly small between the two labeling styles (logistic regression, LQ = 37.58%, LC = 37.42%; K-nearest neighbors, LQ = 44.17%, LC = 44.23%; support vector machine, LQ = 43.36%, LC = 43.47%). This indicates that a small difference in the label accuracy in the training data cannot affect the machine learning performance.

# 7 DISCUSSION

## 7.1 Effects of Labeling Styles in Annotation Efficiency and Label Quality

In psychology, decision-making is a cognitive process in which the cognitive styles of individuals affect the decision-making process as well as the decision outcomes and quality [37] [38] [39]. In manual data annotation, a labeling style can be considered as a decision-making process (i.e., selecting an appropriate label for an image) that may affect the outcomes and quality of the label. In general, the quick labeling style requires less time to complete an annotation task, whereas the careful labeling style requires more time, and it can result in higher quality data. However, it depends on the data difficulty and annotation tasks. Our results indicate that there was no significant difference between the quick and careful labeling styles in the task completion time and label quality when the data were easy (i.e., MNIST). However, there are differences between the two labeling styles when the data becomes difficult. For instance, the careful labeling style requires more time to complete a labeling task that contains moderately difficult images than the quick labeling style. Moreover, it significantly improves the label quality. However, if a labeling task contains extremely difficult images, the careful labeling style cannot improve the label quality and it requires

longer time to complete the labeling task. These results indicate that the labeling style affects the annotation efficiency (task completion time) and label quality (accuracy rate) in non-expert data annotation. However, these effects are dependent on the data difficulty of the annotation task. In addition, the questionnaire results indicated that the subjective impression of the annotator's confidence during annotation was not affected by the labeling styles in any of the labeling tasks with different data ambiguities. However, the confidence of the annotator was affected by the data ambiguities (higher confidence in less difficult data and lower confidence in more difficult data).

## 7.2 Temporal Effects in the Quick and Careful Labeling Tasks

The temporal effect has been used to analyze task performance during image-labeling tasks [16]. It describes ways in which people change their behavior over time, which is a method for analyzing the efficiency of an activity or study [42, 43, 44]. Our results indicated that there was a significant temporal effect (p < 0.05) during the labeling process using the careful labeling style when the images were moderately difficult (Fashion-MNIST). The participants could reduce the task completion time in the second half of the image labeling task by using the careful labeling style. This indicates that the careful labeling style not only improves the label quality but also causes a temporal effect during annotation in a labeling task containing moderately difficult images. Furthermore, there were significant temporal effects (p < 0.05) during annotation using the careful labeling style when the images were extremely difficult (Kuzushiji-MNIST). However, the participants spent longer time completing the second half of the image-labeling task than in the first half. In addition, there was no significant temporal effect (p > 0.05) during annotation using both the quick and careful labeling styles when the images were easy (MNIST). However, the reason for the temporal effect has not been clearly demonstrated.

## 7.3 Effects of Labeling Styles on Machine Learning Performance

The data quality plays a critical role in machine learning. Our user study demonstrated that the careful labeling style can significantly improve the label quality of the image labeling task, which contains moderately difficult images, and it slightly improves the label quality when the task contains extremely difficult images. The machine learning experiment also showed similar results. Labeled data collected via the careful labeling style can result in better machine learning performance (higher accuracy) than that collected via the quick labeling style when the images are moderately difficult (Fashion-MNIST). However, the machine learning performance showed almost no difference between the labeled data collected via the two labeling styles when the images were easy (MNIST), and small differences when the data were extremely difficult (Kuzushiji-MNIST). Machine learning algorithms often work well even if the labels given to difficult data contain errors. This indicates that different label qualities may result in no difference in the machine learning accuracy. In such cases, the labeling style may not be a variable in machine learning (only in manual data annotation). However, our results indicated that the improvement in label quality via the careful labeling style could increase machine learning accuracy when the data are moderately ambiguous. This finding indicates that the careful labeling style can benefit both data annotation and machine learning. However, this depends on data ambiguities. Our machine learning experiment used basic algorithms that only

showed labeling style as a variable in the machine learning performance. Therefore, a machine learning experiment with advanced techniques (e.g., deep learning with a large-scale dataset) still needs to be implemented in the future.

## 7.4 Three Factors for Selecting an Appropriate Labeling Style for an Annotation Task

Our study and machine learning experiment have shown that different labeling styles have their advantages and disadvantages for different annotation tasks. For instance, conducting a careful labeling task is costly (requires longer time to complete a task) than a quick labeling task; however, it cannot guarantee the improvement of the label quality at all levels of data difficulty. Therefore, it is important to decide a reasonable labeling style for an annotation task; otherwise, it may be a waste of time and money if the improvement is not clear. Here, we discuss three factors that should be carefully considered when selecting a labeling style for an annotation task.

(1) Data Difficulty

Our results indicated that data difficulty is a crucial factor affecting annotation results when using different labeling styles. For instance, the careful labeling style only shows benefit (i.e., improves the label quality) when the data is easy and extremely difficult, whereas the quick labeling style only shows benefit (i.e., requires less time) when the data is easy. Based on the results, we suggest that the quick labeling style is a reasonable choice when conducting an easy annotation task. However, when an annotation task contains difficult data, a careful labeling style can be worthwhile.

(2) Annotator Type and Task Conditions

Although this study only focused on non-expert annotators, we believe that the annotator's experience (i.e., domain knowledge in the given task) is an important factor that may significantly affect the annotation results of using different labeling styles. For instance, the data difficulty depends on individual experiences and subjective impressions. We suggest that qualification is important and necessary for recruiting annotators when conducting annotation tasks with different labeling styles. In addition, the task condition (e.g., crowd tasks and in-person tasks) should be considered when deciding the labeling style for an annotation task. Crowdsourcing is a popular approach for conducting annotation tasks, such as Amazon Mechanical Turk [41]. However, the quality of crowd tasks is a critical issue that has been discussed for many years [22, 35]. We believe that this issue also occurs when different labeling styles are used in a crowd task. Therefore, we recruited participants (annotators) for the user study via a professional company. This helps us explore the effects of labeling styles more precisely ( to prove the research concept). However, crowdsourcing remains an indispensable approach for conducting annotation tasks. We suggest that an online workflow should be carefully designed to control the annotation quality, even with different labeling styles.

(3) Instruction for Implementing the Labeling Style

After deciding on the labeling style, it is important to ensure that annotators can follow and implement the labeling style precisely. In this study, we provided textual and oral instructions for each labeling style by an instructor before starting a formal task (including a trial). This is only for a user study. We believe that instructions for using a labeling style are not sufficient in a realistic annotation task. This is because some annotators may be inherently careful to follow an assigned labeling style, whereas others may be inherently sloppy. To avoid this kind of bias, we suggest that a specific labeling workflow (or labeling interface) be designed and provided to afford different labeling styles, or "FORCE" annotators to follow specific steps. For instance, a workflow that requires annotators to double-check or spend a certain amount of time before making a label decision when an annotation task is conducted via the careful labeling style should be designed.

## 8 LIMITATION AND FUTURE WORK

One limitation of this study is that the size of the training data (1200 labeled images collected via each labeling style for each dataset) was small in the machine learning experiment. This is the main reason for the significantly lower machine learning accuracy in our study compared to the benchmarks [24] [25] [26]. Another limitation is that we only used basic machine learning algorithms for training and testing our collected data. However, the main purpose of this study is not to pursue high accuracy of machine learning results, but to focus on the effects of the labeling styles. Our results indicate that the careful labeling style can improve the label accuracy in manual data annotation as well as increase machine learning accuracy. We believe that the labeling styles might have an even greater effect on large-scale labeling tasks and advanced machine learning techniques (e.g., deep learning). In the future, we plan to conduct a large-scale user study via crowdsourcing and test more machine learning algorithms.

Another limitation is the careful labeling style used in this study. In the current instruction (design) of the careful labeling style, the participants were asked to select a label for an image as carefully as possible without time limitations. This condition may be insufficient to conduct a precise label-careful task. A more specific condition or workflow (e.g., allowing modification or force to double-check) for a careful labeling task may be needed for further investigation. In addition to the labeling style, the level of data difficulty should be carefully defined. For instance, how to define the "too easy" and "too difficult" data for each annotator because different annotators may feel different about the same data. In the future, we will explore more details regarding the careful labeling style. For instance, the cause of the temporal effect and the effect of compensation were not clearly demonstrated in this study. Another interesting possibility is the dynamic control of the labeling styles during annotation. If a system can judge the difficulty of each data item before an annotation, it might be possible to ask an annotator to use an appropriate labeling style (using a careful labeling style for data with only moderate difficulty).

## 9 CONCLUSION

In this study, we investigated the effects of labeling style on non-expert data annotation and machine learning. We conducted a user study to compare the quick labeling and careful labeling styles for a manual image annotation task, and we used the labeled data (as training data) to perform a machine learning experiment. Our results indicated that the labeling style is a variable in the data annotation process and machine learning performance. The careful labeling style improves the label accuracy only when the task is moderately difficult, whereas it only increases the cost without improving accuracy when the task is easy or extremely difficult. These findings provide insights for annotators when selecting an appropriate labeling style for an annotation task. This could be an alternative solution for improving non-expert annotations.

**ACKNOWLEDGMENTS**

This work was supported by JST CREST Grant Number JP-MJCR17A1, and JST, ACT-X Grant Number JP-MJAX21AG, Japan.

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
