# OpenReview forum: "An Empirical Study on the Effect of Quick and Careful Labeling Styles in Image Annotation"
_graphicsinterface.org/Graphics_Interface/2022/Conference — GI 2022_

### Official Review · Reviewer_BgDd · 2021-12-21
**An interesting research that is also an important in the field of non-experts data annotation.**

**Rating:** 7
**Confidence:** 5

**Review:**

The paper discusses the human cognitive process in data labelling tasks for later use in machine learning. The authors describe their study in determining two distinctive styles of labelling behaviour and compared the accuracy performance from training the same structured classifiers.
The topic of study is interesting, and yet indeed was not an active field of research even though it plays a significant role in machine learning. The research is therefore novel.
However, there are a few comments and questions which could be addressed in the future.

1. The data labels were certainly in the static number of categories (ten), yet the difficulties in recognizing between F-MNIST versus others could be subjective, especially when some participants seem to be older than the others. Could age play a role in performing the task?
2. In careful labelling condition, it was restricted to make a change once they assign a label to an image. However, since it is ‘careful labelling’, making modifications post-decision could be useful.
3. It was confusing to understand the procedure of data sampling. It says, ‘we randomly selected 100 images (...), then created twelve 100-image datasets’. Were there twelve duplicated sets from the first 100 images? Also, how did you handle the bias between labels?
4. Figure 4 and Figure 5 are identical, one can be removed.
5. It would be useful to know the reasoning behind determining the number of data for testing. For example, why was it 10,000? not 1:2 ratio to the data collected? The initial dataset could also be modified to be a testing/validation set.
6. The t-test results could be reported in detail.
7. Could the temporal effect be understood by human fatigue over time? Labelling 100 images may cause fatigue as approaching the latter set.
8. The machine learning accuracy could further be reported. What about the mean squared error rate? Again, what about the bias between labels?
9. Given that the hourly monetary expense was reported, it can be interesting to see the relationship and trade-off between efficiency to resource spent.

---

### Official Review · Reviewer_CRpa · 2022-01-12
**Well-written, useful and interesting work; needs minor improvement of conclusions and limitations**

**Rating:** 9
**Confidence:** 4

**Review:**

This paper addresses an important topic of designing data labeling tools for accuracy and efficiency, and presents a well-designed, interesting and thorough analysis by studying two labeling styles (quick and careful) and 3 datasets of varying difficulty for 10-class image classification. The dataset splits, labeling task design and participant recruiting are sound, and analysis is thorough, well-presented and insightful. In order to improve the paper, I would suggest improving the conclusions in Sec. 7 and rewriting the limitation sections to consider generality across use cases (see below). The strongest conclusions should be summarized in a prominent location with enough detail to inform practitioners.

**On conclusions:**

6.2: Would these conclusions generalize? How does one decide if a task is “too easy” or “too hard” for careful labeling?

Fig 10: this is really interesting! The time to complete task indicates that labelers were able to learn the labeling task in the case of Fashion-MNIST but not Kuzushiji-MNIST.

7.4 (1): It seems when data is easy, there is no big difference in time (cost) for labeling (Fig. 7), so the conclusion seems to pick quick (intuitive) style only for data that is too difficult. Careful style is either equivalent or brings benefits in other conditions. That is quite an insightful conclusion to draw from this work!

7.4 (2): This is more speculation, not conclusions drawn from this work in particular. Perhaps instead the authors could suggest a pilot to assess difficulty level, given the annotator pool in order to choose labeling style appropriately? For example, one could watch out for the indicator that annotators are able to learn the task (like with Fashion-MNIST) by watching completion time improve temporally.

7.4 (3): It would be instructive to see the distribution of task durations across two labeling styles to substatiate the note that some annotators are naturally “more sloppy”.

**On limitations:**

Section 8: The limitations lisited are minor. This section should be rewritten with broader considerations in mind. The obvious question is: do these findings generalize to other datasets and other labeling tasks (e.g. image segmentation, classification with more than 10 labels, captioning, etc.)? The main limitation of this work is inability to answer the question of how applicable the conclusions are beyond the tasks studied here. This paper does an excellent job at designing the quick and careful labeling tasks with a very general time constraint, which suggests that findings *may* indeed generalize. Please expand on larger implications and future work to design robust guidelines for labeling task design.

**Minor comments:**

4.1 First sentence is incomplete

Fig 5: It seems the wrong image was used (repeat of Fig 4), and the text referrs to a,b,c missing from figure.

4.3, last paragraph: Dataset split description is a little unclear. Were 1200 images selected from each dataset, then split into 12 non-overlapping subsets?

7.4 (3): typo (follow, not fellow)

---

### Official Review · Reviewer_3s5X · 2022-01-14
**mildly in favor if paper can be edited**

**Rating:** 6
**Confidence:** 4

**Review:**


This paper discusses the differences between a "quick" and a "careful" labeling style for image annotation. Machine learning applications benefit from accurate labels; more attention to detail can in principle produce higher accuracy, and this paper seeks in part to characterize the resulting tradeoffs. The authors report experiments contrasting the two labeling styles on easy, medium, and difficult datasets.

The topic is worthwhile and I have not seen this specific division discussed previously. I am in favor of the paper being accepted. However, I have some comments that I hope can improve the paper before publication.

The paper is reasonably well organized. The writing can still be improved, though. Minor spelling errors ("carful" for "careful", "fellow" for "follow") appear throughout. Some passages are unintelligible as written (e.g., "but and some may inherently sloppy"). The paper would benefit from thorough editing.

The machine learning experiment was not very informative. Save the space for further exposition on the user study. There is a substantial literature on ML in the presence of noisy data, which could be drawn on here.

How confident can we be that the participants complied with the requested labeling style? Is "style" even the right word? It suggests something intrinsic to the user, rather than something consciously adopted.

Would people adapt naturally to the dataset difficulty? It seems so from the results in figure 7. Can one predict the dataset difficulty ahead of time? If the paper's results are correct, there is no advantage to a "careful" style for very difficult datasets; does this result lead to actionable advice?

The subjects did not see the same data, as far as I can tell. Is this really between-subjects, then? If the subjects actually saw the same data items, it would be possible to do more analysis, identifying data that was genuinely difficult as opposed to simple data where errors are more likely the results of misclicks than confusion. All elements in a given dataset are probably not equally difficult.

Minor points:

It would be better to provide a report of per-label time, rather than total time to label everything. By reporting only totals, the distributions have been changed, since averaging over many data points has already destroyed some data.

I am not clear on the ethics of outsourcing the experiment in the manner described. Usually participants would be recruited in a fashion that limited the possibility of coercion, a presumptive possibility when the participants' employer requested that they join the experiment.

I have some doubts about the users' reported confidence, since it was measured retrospectively over an entire dataset. Users do not have the cognitive capacity to accurately estimate their average confidence over a large number of actions. It would be better to report per-label confidence. If this is too onerous, the system could poll a subset of labels, e.g., every fifth label.

While the "quick" and "careful" styles have clear intuitive meanings, the paper mixes in the concepts of "intuitive" and "systematic" decisionmaking in an ambiguous way. The users in the "careful" style took longer, but nothing suggests that they necessarily took a systematic approach.

---

### Decision · Program_Chairs · 2022-01-18

Accept